# Pre-Drying of Chlorine–Organic-Contaminated Soil in a Rotary Dryer for Energy Efficient Thermal Remediation

**DOI:** 10.3390/ijerph192416607

**Published:** 2022-12-10

**Authors:** Rui Chai, Jinqing Wang, Mingxiu Zhan, Dingkun Yuan, Zuohe Chi, Hailin Gu, Jiani Mao

**Affiliations:** College of Metrology and Measurement Engineering, China Jiliang University, Hangzhou 310018, China

**Keywords:** soil pre-drying, non-isothermal drying, indirect rotary dryer, pollutant desorption, volumetric heat transfer coefficient

## Abstract

In response to the current problem of the high energy consumption of direct thermal desorption systems when treating soils with a high moisture content, we propose using the waste heat of the system to pre-dry soil to reduce its moisture. Taking chlorine–organic-contaminated soil as an object, an experimental study on the drying and pollutant desorption characteristics of soil in an indirect rotary dryer was carried out. The results show that the non-isothermal drying process was divided into warm-up and falling rate periods, and no constant period was observed. The higher the rotation speed, the lower the soil outlet temperature and the higher the drying tail gas temperature. Soil outlet and dry tail gas temperatures were lower for soils with a higher moisture content. Benzene and cis-1,2-dichloroethylene are easily desorbed. Therefore, the disposal of dry tail gas should be determined according to the type and concentration of soil pollutants present. The volumetric heat transfer coefficient was found to be 85–100 W m^−3^ °C^−1^, which provides a key parameter for the size design of a rotary dryer.

## 1. Introduction

With the rapid development of industrialisation in China and the increasing awareness of the importance of environmental protection among the public, polluting enterprises are gradually being renovated or relocated, resulting in a large number of legacy pollution sites. These sites are dominated by organic pollutants characterised by a wide range of species, high concentrations and uneven distribution. Common remediation methods for treating organic soil include soil leaching [1], the microbial method [2], direct thermal desorption [3,4] and so on. Direct thermal desorption, a widely used technology for the remediation of organic-contaminated sites, can quickly and efficiently remove volatile and semi-volatile organic compounds, such as PAHs [5], PCBs [6] and total petroleum hydrocarbons [7]. Its advantages include the complete removal of contaminants, short remediation cycles and flexible treatment sites [8,9].

Organic-contaminated soils usually have a high moisture content. During soil thermal desorption, the moisture in the soil evaporates and is sent to a secondary combustion chamber together with the harmful gas generated. The combustion temperature of the secondary combustion chamber is above 900 °C; most of the organic pollutants in the exhaust gas will burn out at this temperature [10]. In this process, water vapor will also be heated to 900 °C, resulting in energy waste. Troxler et al. [11] determined that when the soil moisture content is 10–15%, the energy required to heat the soil is nearly the same as that required to evaporate the water. Xu et al. [10] analysed the energy consumption of a thermal desorption system. The obtained data showed that reducing the soil moisture from 20% to 15% can reduce the energy consumption by more than 20%. Therefore, this study examined the effect of a pre-drying step before thermal desorption on the reduction in the soil moisture content. As shown in Figure 1, the pre-drying energy can be obtained by calculating the waste heat of the flue gas emitted by the thermal desorption system, which would imply a significant reduction in the energy consumption of the system.

The key to efficient removal of water from soil lies in the design of the dryer and the drying characteristics. Rotary dryers are often used to dry agricultural products, such as grains and coffee [12], as well as industrial materials, such as cement and sludge [13], especially in the field of sludge drying. Most studies on rotary sludge drying have focused on two particular aspects: rotation speed and sludge moisture content. Wen et al. [14] investigated the effect of cylinder speed on the evaporation intensity and evaporation coefficient of the dryer, as these parameters are closely related to the drying effect. Deng et al. [15] investigated the effect of the rotation speed of the sludge drum dryer on the drying effect. They found that increasing the rotation speed improved the drying efficiency, but only to a certain extent; too high a speed could even have a negative effect on the drying process. Poós et al. [16] experimentally studied the morphology of sludge with different moisture contents in a rotary dryer and used a re-mixing technique to reduce the moisture content of the inlet sludge and found that the sludge gradually granulated when the moisture content was reduced to 65%. Hamawand et al. [17] studied the effect of moisture content on sludge stickiness and agglomeration in a drum dryer. They found that mixing the sludge with lignite significantly reduced both the stickiness and agglomeration of the sludge and led to a significant reduction in the moisture content of the final product. Organic-contaminated soils have a lower moisture content than sludge, and soil particles pretreated by crushing and screening are uniform in size, stable in shape and present good air permeability, so they are more suitable for treatment in a rotary dryer. However, little research has been conducted on the drying characteristics and pollutant desorption of organic-contaminated soil in rotary driers, and thus there is a lack of corresponding reference values for the design of the main thermal parameters, restricting the application of this technology for thermal desorption.

In this study, the non-isothermal drying characteristics of an organic-contaminated soil was investigated using a thermogravimetric (TG) method. Thermogravimetric–mass spectrometry (TG–MS) was used to explore the relationship between pollutant desorption in the soil and drying temperature. Moreover, a small test platform for an indirect rotary dryer with a processing capacity of 50 kg h^−1^ was constructed to study the effect of rotation speed and soil moisture content on drying characteristics. The desorption of pollutants during the drying process was analysed, and the reference values of the main parameters of the thermal design of the rotary dryer were calculated.

## 2. Materials and Methods

### 2.1. Experimental Materials

The organic-contaminated soil used in this study was obtained from a chlorine-organic-contaminated site in a pesticide factory in China. The initial components and concentrations of the soil pollutants were determined using gas chromatography–mass spectrometry, and the results are shown in Table 1. The average mass moisture content (9.21%) of the initial soil sample was determined using the weight loss method. To verify the influence of moisture on the drying process, different amounts of water were added to two groups of soil samples, and these were allowed to stand for 24 h, after which the moisture was measured again. The obtained average mass moisture content of the two groups was 14.58% and 22.06%, and the particle size of the pretreated soil ranged from 5 to 8 mm.

### 2.2. Experimental System

The experimental system, as shown in Figure 2, had a soil treatment capacity of 50 kg h^−1^. To heat the cylindrical body, the high-temperature flue gas from the secondary chamber entered the spacer layer between the rotating drum and the insulation layer. The direction of the flue gas flow was consistent with that of the soil movement in the rotary dryer. Thermocouples were installed at the flue gas inlet and outlet to measure the temperature. T1 was located at the outlet soil, where a J-type thermocouple was installed to measure the soil temperature, whereas Y1 was located in the center of the dry tail gas outlet, and it was equipped with a J-type thermocouple to measure the temperature of the dry tail gas. The J-type thermocouples installed at T1 and Y1 were connected to a data acquisition instrument (KEYSIGHT type 34927A), which was used to read the temperature values. After leaving the dryer, the dry tail gas entered the bag filter and, after dust removal, was condensed, after which it entered the chimney through the pipeline for discharge. A sampling port was arranged in front of the chimney on the pipe for the collection of dry tail gas samples.

An indirect rotary dryer was used in the experiment (Figure 3). During the treatment, the soil was passed through the feeding, heating, and discharging spirals, in that order. The heating section included a built-in L-shaped lifting plate.

### 2.3. Experimental Method

TG STA 449 F5 (NETZSCH) was used to conduct non-isothermal drying experiments on soil samples so as to identify their drying characteristics. The mass of each group of samples was 100 ± 1 mg, and the heating rates were 5, 10, 15, and 20 °C min^−1^. Considering that the on-site drying temperature is generally below 200 °C, the temperature range for the drying experiment was set to 30–200 °C. The pollutant release characteristics of the soil samples were investigated using TG–MS (STA 449 F3, QMS 403). To obtain the release curve of pollutants, the heating range of TG–MS was set to 30–350 °C, and the heating rate was set to 10 °C min^−1^.

The effects of rotation speed and soil moisture on the drying process were investigated, and the working conditions were set as shown in Table 2. In the experiment, the inclination angle of the rotary dryer was 2°, and the heat was sourced from the high-temperature flue gas of the secondary combustion chamber. The temperature of the inlet flue gas was stable, at 745–750 °C. Contaminated soil was continuously added to the rotary dryer at 50 kg h^−1^, and the dryer was run for 30 min until the drying device ran stably. The outlet soil temperature and drying tail gas temperature were then measured. The soil at the dryer outlet was sampled to measure the soil moisture after drying. The dry tail gas at the chimney was collected under the L3 operating conditions to investigate its pollution potential. The concentration of each pollutant in the dry tail gas was detected offline using Fourier transform infrared spectroscopy. To ensure reproducibility, the experiments were repeated three times.

## 3. Results and Discussion

### 3.1. Soil Drying Characteristics

Figure 4 shows the TG–DTG (differential thermogravimetry) curves of soil samples (moisture content of 9.21%) dried at four different heating rates. The figure shows that throughout the non-isothermal drying process, there were obvious warm-up and falling rate periods, and no constant rate period was observed, as similarly seen for sludge drying [18]. The critical point temperature (the warm-up and falling rate periods’ turning point temperature) [19] and the corresponding moisture content are indicated in the diagram. Sufficient moisture was observed on the soil surface during the warm-up period, and a large amount of water evaporated during this period. By the critical point, the percentage of dried soil moisture at the four different heating rates, 5, 10, 15 and 20 °C min^−1^, was 71.5%, 73.4%, 73% and 71.1%, respectively. After the critical point, the rate of evaporation of soil moisture depends mainly on the diffusion of water within the soil [20], at which point more energy is required to evaporate the moisture. 

Since most of the moisture in the soil is dried during the warm-up period, the critical point temperature is a suitable drying temperature for reference. The critical point temperatures for chlorine–organic-contaminated soils in this study were 61.1 °C, 73.6 °C, 82.3 °C and 92.2 °C at the four different heating rates. For the same drying equipment, in the case where the drying requirements are met, the lower the soil outlet temperature is, the less energy is consumed; however, lower soil outlet temperatures are not necessarily better, as they indicate a slow soil heating rate, which means the rotation speed of the rotary dryer must be decreased, directly reducing the processing capacity of the drying equipment. The above process should thus be considered in engineering practice.

### 3.2. Influence of Rotation Speed on Drying Process

Table 3 shows the drying parameters of the soil at different rotation speeds. The slower the rotation speed, the more time needed for the soil to be heated in the dryer, the higher the soil outlet temperature and the lower the soil outlet moisture content [13]. In contrast to the increase in soil outlet temperature, the drying tail gas temperature decreased with decreasing rotation speed, from 110.8 °C at 3 r min^−1^ to 70.3 °C at 1 r min^−1^. This occurred because lower rotation speeds led to weaker turbulence of the flue gas outside the inner cylinder, which led to a lower heating effect of the flue gas on the inner cylinder wall. In addition, when the rotation speed was low, the ability of the lifting plate to lift the soil weakened, and at a certain feed rate, a large amount of soil accumulated on the inner wall of the rotary dryer, thereby decreasing the heat exchange effect between the dry tail gas and the dryer. 

The soil heating rates for L1, L2 and L3 were 5.98, 8.63 and 11.83 °C min^−1^, respectively, which are similar values to those observed for the 5 and 10 °C min^−1^ heating processes for the non-isothermal drying test. In comparing the TG-DTG test results in Figure 4, it was found that the soil thermogravimetric moisture content largely matched the soil drying experimental outlet moisture content at the same soil outlet temperature (see Table 3), indicating that the soil was in a falling rate period at the outlet of the dryer in all three working conditions. It can be seen that the results of the TG-DTG test of the soil are a good predictor of the drying stage of the soil in the dryer.

### 3.3. Influence of Moisture Content on Drying Process

The outlet temperatures of the chlorine-organic-contaminated soil and the temperatures of the dry tail gas under different moisture contents are shown in Figure 5. The increase in soil moisture led to a decrease in the temperature of outlet soil and dry tail gas. This occurred because the specific heat capacity of water is significantly larger than that of soil, and water has a larger latent heat (required for evaporation). Therefore, under stable dryer operation, the heat provided was nearly unchanged, and the heating and evaporation of water in the high-moisture soil thus consumed more energy, which decreased the temperatures of the outlet soil and dry tail gas.

The average mass moisture of the soil measured after drying under working conditions L3, M3 and H3 was 0.25%, 0.32% and 0.19%, respectively. That is, despite the significant increase in soil moisture, the moisture of the outlet soil was low, which indicates that the rotary dryer is efficient for the treatment of high moisture soil at 3 rpm.

For the dry tail gas, water vapor condensation occurred in the bag filter under working conditions M3 and H3 due to the increase in soil moisture, which led to an increase in the humidity of the dry tail gas [21]. Moreover, as the temperature of the dry tail gas decreased, its water vapor content was more likely to condense and separate. Therefore, in actual practice, the temperature of the dry tail gas at the outlet of the dryer should be appropriately increased to above the dew point according to the distance between the dryer and the bag filter. 

### 3.4. Desorption of Pollutants in Contaminated Soil during Drying Process

The soil samples (moisture 9.21%) were analysed using TG–MS, and the measured ion current intensity curve of the main pollutants is shown in Figure 6. The content of benzene in the dry tail gas began to increase after the temperature reached 65.32 °C, and it reached a peak at 252 °C, after which it gradually decreased. The content of cis-1,2-dichloroethylene began to increase after 58.5 °C, but no peak appeared within 30–350 °C. Several main organic pollutants started to desorb at temperatures near the boiling temperature (the parentheses in Figure 6 represent the boiling temperature of the substance) during the soil drying process, but a sharp increase in content occurred after the drying temperature increased to 156.5 °C, which was higher than the temperature experienced by the soil in the dryer. 

The dry tail gas of working condition L3 was evaluated offline, and the results are shown in Table 4. Benzene and cis-1,2-dichloroethylene presented the highest concentrations, whereas other pollutants showed relatively low concentrations. The chloroform concentration in the soil samples reached 25.3 mg kg^−1^, and it was not detected in the dry tail gas. Based on Figure 6, the drying temperature of the soil in the rotary dryer was lower than 156.5 °C, and chloroform was not desorbed in large quantities at that point. The initial content of ethylbenzene in the soil was low, as was the concentration detected in the dry tail gas. The dichloromethane concentration in the initial soil sample was 23.4 mg kg^−1^, whereas its concentration in the dry tail gas was less than 0.3 mg m^−3^. This might be attributed to the low boiling point of dichloromethane (39.8 °C), which likely thermally escaped during on-site storage and mechanised transportation.

In summary, secondary pollution might be caused by the desorption of pollutants during the drying of contaminated soil. Owing to the easy desorption of benzene and cis-1,2-dichloroethylene, the dry tail gas can be condensed to remove water before being introduced into the secondary combustion chamber for combustion treatment. Alternatively, adsorption or catalytic oxidation can be used to remove the organic matter from the tail gas.

### 3.5. Volumetric Heat Transfer Coefficient

The volumetric heat transfer coefficient is an important parameter used to measure the heat transfer performance of industrial drying equipment. In the design of a rotary dryer, its geometry is usually determined by the selected volumetric heat transfer coefficient. Therefore, it is of great significance to determine the volumetric heat transfer coefficient of rotary drying equipment used for the treatment of organic-contaminated soil. 

The volumetric heat transfer coefficient is defined as the heat transferred in the unit volume of the drying equipment per unit of time under the unit heat transfer temperature difference [22]:(1)$$K_{V}=\frac{Q}{V\cdot \Delta T_{m}}$$
where $K_{V}$ is the volumetric heat transfer coefficient, W m^−3^ °C^−1^; Q is the total heat transfer in the rotary dryer per unit of time, *W*; V is the effective volume of the rotary dryer, m^3^; and $\Delta T_{m}$ is the logarithmic mean temperature difference of the heat transfer, °C.

The total heat transfer  Q can be expressed as:(2)$$Q=Q_{w}+Q_{s}+Q_{l}$$
where $Q_{w}$ is the heat required to dry the material, *W*; $Q_{s}$ is the heat absorbed by the moisture of the material, including the sensible heat (temperature rise) and the latent heat (vaporisation), *W*; and $Q_{l}$ is the heat required to increase the temperature of the drying medium from the inlet to the outlet, *W*.

The effective volume of the rotary dryer can be calculated as follows:(3)$$V=\frac{\pi }{4}D^{2}L$$
where D and L are the effective diameter and heating length of the rotary dryer, respectively, m.

The logarithmic mean temperature difference for heat transfer is calculated as:(4)$$\Delta T_{m}=\frac{\left(t_{1}-t_{m1}\right)-\left(t_{2}-t_{m2}\right)}{ln[\left(t_{1}-t_{m1}\right)/\left(t_{2}-t_{m2}\right)]}$$
where $t_{1}$ and $t_{2}$ are the inlet and outlet flue gas temperatures, respectively, °C; $t_{m1}$ and $t_{m2}$ are the inlet and outlet material temperatures, respectively, °C.

The volumetric heat transfer coefficients of working conditions L3, M3 and H3 were calculated, and the results are summarized in Table 5. The volumetric heat transfer coefficient of soil under different moisture contents ranged from 85 to 100 W m^−3^ °C^−1^. In engineering practice, this range could be used to select the volumetric heat transfer coefficient of an indirect rotary dryer. To ensure the drying of contaminated soil with a high moisture content, the smaller value within this range should be used.

## 4. Conclusions

(1)The non-isothermal process of drying organic-polluted soil can be divided into warm-up and falling rate periods, and no constant rate period was observed in soil drying. More than 70% of the moisture evaporation in the soil was concentrated in the warm-up period. The critical point temperature is used for the selection of the drying temperature of a rotary dryer. When the drying requirements and processing efficiency are met, a low critical temperature can be selected to reduce the energy consumption of the system.(2)A low rotation speed has a greater effect on the drying tail gas temperature and a smaller effect on the outlet soil temperature. In the case of measuring the soil outlet temperature and moisture content, the drying stage of the soil in the dryer can be evaluated using the soil thermogravimetric test results.(3)A high soil moisture content will increase the humidity of the dry tail gas while lowering the temperature of the dry tail gas and making it easier for the moisture in the dry tail gas to condense and precipitate. In engineering practice, the temperature of the dry tail gas should be increased to above the dew point.(4)Soil containing benzene and cis-1,2-dichloroethylene is easily desorbed during the drying process. Therefore, in engineering practice, the dry tail gas disposal scheme should be designed according to the type and concentration of soil pollutants to avoid secondary pollution.(5)When using an indirect rotary dryer to treat organic-contaminated soil, 85–100 W m^−3^ °C^−1^ may be considered the reference range for the volumetric heat transfer coefficient.(6)Shortcomings of this article include a low rotary dryer rotation speed and the fact that the influence of a high rotation speed on the soil drying effect is not explored. The critical rotation speed was not obtained; that is, at this speed, the soil drying requirements can be met, and the processing capacity of the equipment can be maximised.

## Figures and Tables

**Figure 1 ijerph-19-16607-f001:**
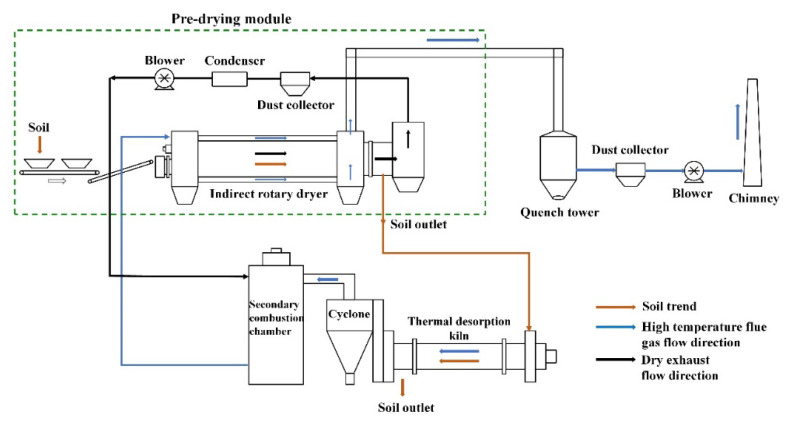
Direct thermal desorption system with pre-drying step to treat organic-contaminated soil.

**Figure 2 ijerph-19-16607-f002:**
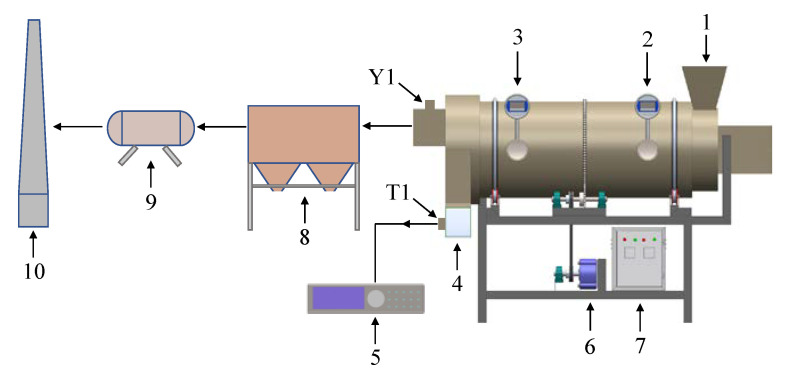
Schematic diagram of pre-drying experimental system: (1) inlet soil, (2) flue gas inlet and gas temperature indicator, (3) flue gas outlet and gas temperature indicator, (4) soil export, (5) data acquisition instrument, (6) electric machinery, (7) electric cabinet, (8) bag filter, (9) condenser and (10) chimney, T1/Y1: sampling hole.

**Figure 3 ijerph-19-16607-f003:**
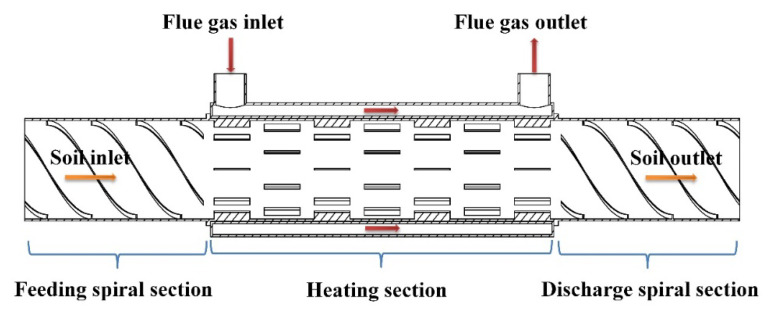
Schematic diagram of internal structure of small rotary dryer.

**Figure 4 ijerph-19-16607-f004:**
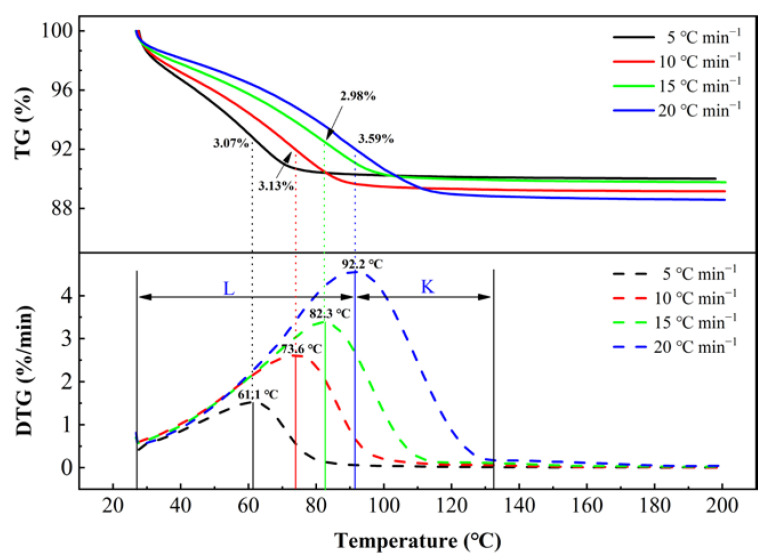
TG and DTG curves of soil samples dried at different heating rates of 5, 10, 15, and 20 °C min^−1^ (20 °C min^−1^ is used to illustrate the segmentation, where L = warm-up period; and K = falling rate period).

**Figure 5 ijerph-19-16607-f005:**
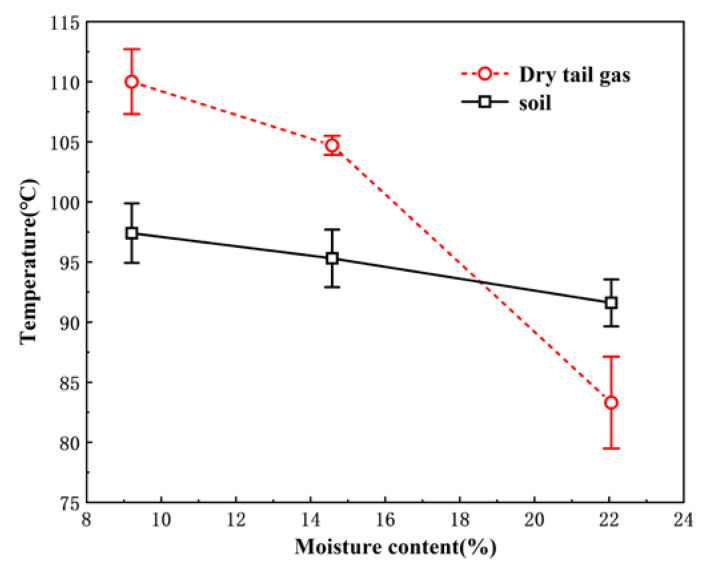
Temperature map of soil and dry tail gas from the rotary dryer for different moisture contents.

**Figure 6 ijerph-19-16607-f006:**
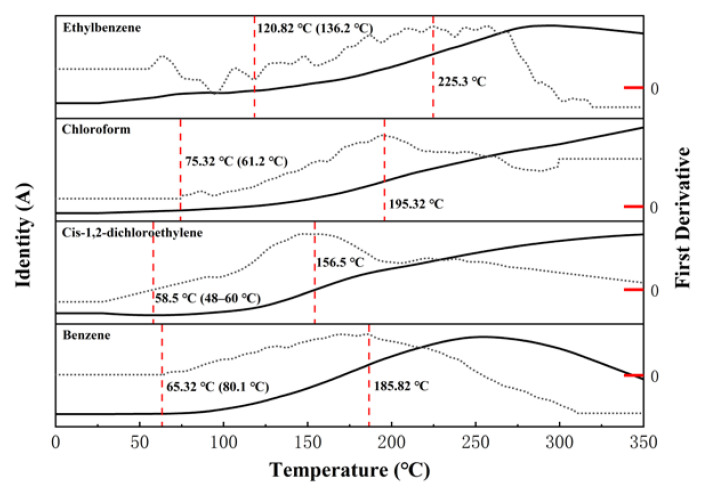
Ion current intensity curve of the main organic pollutants desorbed during the drying of the initial soil sample.

**Table 1 ijerph-19-16607-t001:** Concentration of primary pollutants in the soil.

Pollutant	Concentration (mg kg^−1^)
Dichloromethane	23.4
Trans-1,2-dichloroethylene	0.825
Cis-1,2-dichloroethylene	7.04
Chloroform	25.3
1,2-Dichloroethane	1.1
Benzene	0.014
Trichloroethylene	0.088
1,1,2-Trichloroethane	0.155
Tetrachloroethylene	0.173
Ethyl benzene	0.061
M-xylene & P-xylene	0.436
O-xylene	0.218
1,1,2,2-tetrachloroethane	0.921

**Table 2 ijerph-19-16607-t002:** Rotary dryer operating conditions.

Working Condition	Rotation Speed (rpm)	Moisture Content (%)
L1	1	9.21
L2	2	9.21
L3	3	9.21
M3	3	14.58
H3	3	22.06

**Table 3 ijerph-19-16607-t003:** Drying process parameters of soil at different rotation speeds.

Working Condition	T1	T2	t	Tr	T3	w	w_1_
	(°C)	(°C)	(min)	(°C min^−1^)	(°C)	(%)	(%)
L1	10.2	105.5	15′56′′	5.98	70.3	0.11	0.24
L2	10.2	101.5	10′34′′	8.63	94.6	0.14	0.36
L3	10.2	97.4	7′22′′	11.83	110.8	0.25	0.38

T1: average temperature of the inlet soil; T2: average temperature of the outlet soil; t: average dwell time; Tr: soil heating rate; T3: average temperature of dry tail gas; w: average moisture content of outlet soil; w_1_: soil thermogravimetric moisture content corresponding to the average soil outlet temperature.

**Table 4 ijerph-19-16607-t004:** Concentration of pollutants in dry tail gas.

Pollutant	Concentration (mg m^−3^)
Dichloromethane	<0.3
Trans 1,2-dichloroethylene	0.058
Cis-1,2-dichloroethylene	1.03
1,2-Dichloroethane	0.179
Benzene	2.23
Ethyl benzene	<0.010
1,1,2-Trichloroethane	<0.0004
Tetrachloroethylene	0.001
O-xylene	<0.010
Meta-xylene	<0.010
Paraxylene	<0.010
1,1,2,2-tetrachloroethane	<0.00007
Volatile organic compounds	0.139

**Table 5 ijerph-19-16607-t005:** Volumetric heat transfer coefficient for each working condition.

Working Condition	Moisture Content	Volumetric Heat Transfer Coefficient
	**(%)**	**(W m^−3^ °C^−1^)**
L3	9.21	86.56
M3	14.58	92.53
H3	22.06	97.46

## Data Availability

Some or all data that support the findings of this study are available from the corresponding author upon reasonable request.

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
