# Peer review of "Pre-Drying of Chlorine–Organic-Contaminated Soil in a Rotary Dryer for Energy Efficient Thermal Remediation"

_ijerph, 2022, doi:10.3390/ijerph192416607_

Round 1
Reviewer 1 Report
The authors present and interesting and comprehensive analysis of the use of flue gas as a heat source to pre dry contaminated soil prior to the removal of organics by thermal desorption.
Overall, the study is well designed and well written. A few areas that may improve the text are described below.
Line 24: I am not sure what is meant by "a large number of enterprises have been created." Please clarify.
Line 42: How is the water heated to 900°C?
Line 113: Please indicate that the soil is contaminated with chlorinated organics not chlorine.
Throughout - the authors describe precipitation of the organic contaminants as an issue. I am not fully sure what is meant by this. To me, precipitate means either the condensation and deposition of solid and fluid droplets, or the formation of solid particles from a solution. Do the authors mean desorption? Please clarify.
Reviewer 2 Report
This manuscript reports the pre-drying effect of chlorine–organic-contaminated soil in a rotary dryer. There have been many previous studies on the drying of soil or sludge. I don't think this article has good novelty. In the Results and discussion of this manuscript, most of them are just descriptions of the results, lacking in-depth discussion. For example, I don't see any comparison with other research results. In this manuscript I have not seen the results with statistical characteristics presented, resulting in the reliability of the conclusions questionable. In addition, Part 2 is suggested to place in Part Materials and Methods. Pay attention to the difference between conclusions and results. The conclusions of this manuscript needs to be rewritten. I do not recommend that this manuscript be accepted.
Reviewer 3 Report
This manuscript evaluates that efficient remediation of high-moisture organically polluted soils using an indirect rotary dryer coupled of waste heat. Meanwhile, the optimum condition of the device and the harm of the exhaust gas are discussed. This research has important practical significance and application value for the green development of energy conservation and emission reduction.
However, I am not convinced that the paper in its current state is of sufficient quality for publication in the journal of IJERPH. First, the structure of the article is chaotic, e.g. Line 97. It's confusing to put this paragraph here. Perhaps it is appropriate to put it in the section of introduction or methods. Second, the paper is not well written, and it takes quite some time to fully understand the methodology and discussions. Therefore, a major revision in the language and structure of the paper is required. The use of an editing service is highly recommended.
Round 2
Reviewer 2 Report
The manuscript has been greatly revised, and I agree that the current form is accepted
Reviewer 3 Report
After careful revision of the manuscript according to the reviewer's advice, the quality of the manuscript has been greatly improved and has reached the requirements of the journal. It is recommended to receive it.